# Human Gait Activity Recognition Machine Learning Methods

**DOI:** 10.3390/s23020745

**Published:** 2023-01-09

**Authors:** Jan Slemenšek, Iztok Fister, Jelka Geršak, Božidar Bratina, Vesna Marija van Midden, Zvezdan Pirtošek, Riko Šafarič

**Affiliations:** 1Faculty of Mechanical Engineering, University of Maribor, 2000 Maribor, Slovenia; 2Faculty of Electrical Engineering and Computer Science, University of Maribor, 2000 Maribor, Slovenia; 3Department of Neurology, University Clinical Centre, 1000 Ljubljana, Slovenia

**Keywords:** human gait, activity recognition, wearable, machine learning, convolutional neural network, recurrent neural network, attention mechanism

## Abstract

Human gait activity recognition is an emerging field of motion analysis that can be applied in various application domains. One of the most attractive applications includes monitoring of gait disorder patients, tracking their disease progression and the modification/evaluation of drugs. This paper proposes a robust, wearable gait motion data acquisition system that allows either the classification of recorded gait data into desirable activities or the identification of common risk factors, thus enhancing the subject’s quality of life. Gait motion information was acquired using accelerometers and gyroscopes mounted on the lower limbs, where the sensors were exposed to inertial forces during gait. Additionally, leg muscle activity was measured using strain gauge sensors. As a matter of fact, we wanted to identify different gait activities within each gait recording by utilizing Machine Learning algorithms. In line with this, various Machine Learning methods were tested and compared to establish the best-performing algorithm for the classification of the recorded gait information. The combination of attention-based convolutional and recurrent neural networks algorithms outperformed the other tested algorithms and was individually tested further on the datasets of five subjects and delivered the following averaged results of classification: 98.9% accuracy, 96.8% precision, 97.8% sensitivity, 99.1% specificity and 97.3% F1-score. Moreover, the algorithm’s robustness was also verified with the successful detection of freezing gait episodes in a Parkinson’s disease patient. The results of this study indicate a feasible gait event classification method capable of complete algorithm personalization.

## 1. Introduction

Human locomotion is an extraordinarily complex translational motion that can be described on the basis of the kinematics and muscular activity of the extremities in all their various movements [1]. The analysis of human gait identifies and analyzes walking and posture problems, load anomalies and muscle failure, which would not be measurable with normal clinical exams. It can be defined as a pattern of the locomotion characteristic of a limited range of speeds and serves as an important diagnostic tool in many fields, such as health, sports, daily life activities, prosthetics, and can thus have a significant impact on the person’s Quality of Life (QOL) [2].

The analysis of human gait cyclic motion is usually obtained from identification and characterization of an individual’s walking pattern and kinematics. The gait cycle observation can be derived from sensor data approaches (visual, weight, inertial, etc.) and modeled upon different data driven methods. The extracted features (phases) recognized in a gait analysis can be used for many tasks, including, among others, addressing health-based issues, such as recognition of unpredictable gait disorders, which, in the worst cases, can lead to injuries [3,4]. However, two things are important for reliable gait analysis: An appropriate wearable gait motion data acquisition system (with sensors) and methods running for robust gait monitoring, analysis and recognition. A wearable system with sensors, a processing unit and communication, packed in a small lightweight housing, is usually desired for long-term daily use. The important criteria for a wearable system are also low power consumption, sufficient external memory storage, and a user-friendly human interface for on-line data visualization and monitoring. In the case of Machine Learning (ML) methods, many of them now enable feature extraction. A combination of different learning methods can be used to recognize and extract human walking features adequately. A tradeoff between code complexity and real-time operation needs to be considered as far as implementation is concerned. The human gait is a cyclical process, with the joint operation of bones, muscles and the nervous system trying to maintain static and dynamic balance of an upright body position while in motion. The cycle can be defined as the time between two consecutive contacts of the foot striking the ground while walking, and has two states, namely stance when the foot is firmly on the ground, and swing when the foot is lifted from the ground, as seen in Figure 1.

Furthermore, gait phases can be divided into durations of certain events (left and right foot up/down, limb stance/swing, and limb turn) [5], and classified according to the area of interest (sport, health, prosthetics, and daily life) [6,7,8], type of sensors used (floor sensors, visual, inertial or other sensors) [9,10], and last but not least, its placement (shoe, ankle, limb, and hip/back) [11,12]. The gait analysis in sports can improve walking/running techniques, prevent long-term hip, knee and ankle injuries, improve sport results, and aid the development of custom-made athlete’s textiles or sport shoes [6]. In medical applications, a gait analysis can result in successful recognition of various gait disorders, like Freezing Of Gait (FOG) in Parkinson’s Disease (PD) or Cerebral Palsy (CP) patients [3]. In prosthetics applications, the gait analysis can help disabled patients after limb amputation to improve feedback in walking with a robotic prosthesis [8,13]. 

Collecting data of human activities became very accessible with technology development and hardware minimization. Older approaches using visual-based systems (e.g., cameras) [10], and environmental sensors for data acquisition, are well established in applications with remote CPU processing, but are limited to mostly indoor solutions. Approaches based on force (weight) measurements are conducted in a motion analysis laboratory with force platforms (floor tiles) and optical motion systems [14]. Such motion capture systems are not easily portable, and usually only operate in controlled environments. Force-based systems, such as foot switches or force-sensitive resistors, are generally considered the gold standard for detecting gait events, yet they are prone to mechanical failure, unreliable due to weight shifting, and provide no details about the swing phase. 

Small and lightweight unobtrusive devices have become indispensable in human daily life activities. Many research groups and authors have presented their achievements in the field using numerous hardware systems and methods intended for indoor and outdoor use [15]. In the past decade, Inertial Measurement Unit (IMUs) sensors with gyroscopes and accelerometers have become very affordable and appropriate, due to their small-size, low power consumption, and fast processing and near instantaneous accelerations measurements during motion [16,17,18]. Many related research studies on gait analysis are oriented towards using wearable IMU-based sensor devices [7,11,19]. Today, IMU sensors can be found in many commercially available gadgets (i.e., fitness bands, smart watches, phones) with user-friendly User Interfaces (UI) [20]. Usually, embedded device boards are used for dedicated tasks. An important aspect is also sensor placement on the human limb/body, where accelerometer locations for the gait research studies were almost evenly distributed among the shank/ankle [11,13,17,21], foot [5,22,23], and the waist/lower back/pelvis [12,24].

When reviewing the related work for gait analysis, we found that some authors used a foot displacement sensor in combination with IMUs to classify heel strike, heel off, toe off and mid-swing gait phases under different walking speeds, as described in [25]. Most novel methods are based on ML, which is a very popular collection of data-driven methods that take advantage of a large amount of data and reduce the need to create meaningful features to perform the classification task. Various ML approaches, such as K-Nearest Neighbors (KNN) [26], Decision Trees (DT) [27], Naïve Bayesian classifier (NB) [28], Linear Discriminant Analysis (LDA) [10] and Long-Short Term Memory (LSTM) Neural Networks (NN) [29], have been used for gait disorder’s recognition and to distinguish between gait phases. Some authors propose the Hidden Markov Model (HMM) method [8,17,30], or Artificial Neural Networks (ANN), which are largely applied in feature extraction and image recognition and have also been used successfully to identify human motion and activity [13,22,31]. The Support Vector Machines (SVM) method was used for analysis and classification of activities such as running, jumping and walking [32]. Newer studies also involve using electromyography data combined with inertial signals [3], approaches based on higher-order-statistics IMUs and transformation of acceleration signals into features by employing higher-order cumulants [33], and advanced NN models [13,22]. Recent studies outperformed other state-of-the-art methods by utilizing ML with an added attention mechanism. The original article on attention [34] describes the attention mechanism as a function which can learn the relative importance of input features for a given task. The attention mechanism is parameterized by a set of weights, which are learned during training. The hybrid Convolutional Neural Network (CNN) [13,35,36,37] and Recurrent Neural Network (RNN) [38,39] architecture with an added attention mechanism [40,41] can be applied to boost the classification performance of the algorithm further.

The motivation of the paper is to develop a robust and lightweight system capable of human gait activity monitoring, analysis and recognition, with the aim to improve a subject’s QOL if applied in the fields of medicine, sports, or prosthetics. The proposed system (Figure 2) consists of the following components: A gait motion data acquisition system, time-series data collection, data pre-processing and data processing (classification). 

The gait motion data acquisition system allows for semi-automatic labeling of recorded gait data, needed later during the learning phase of the ML methods. The purpose of the data collection is to gather data acquired from the gait motion data acquisition system into a database saved on a personal computer. Data pre-processing captures three methods: Feature normalization, feature transformation, and feature reduction. Feature normalization means scaling the parameter values into specific intervals. The feature transformation refers to frequency transformation techniques, where the Continuous Wavelet Transform (CWT) is used typically [42,43]. This method increases the data dimensionality. Data can be processed further using feature reduction methods such as Auto Encoder (AE), which saturates the data and decreases their dimensionality [44]. 

The gait activity was recognized by various ML methods, where the combination of the CNN and RNN algorithms with added attention (CNNA + RNN) was distinguished by the best classification results according to standard classification measures on the RAW, as well as CWT datasets. The two types of datasets were used in order to show the effects of the pre-processing. The robustness of the proposed system using the CNNA + RNN classification method was justified by an experiment, in which five subjects of different ages and gender were included. Obviously, each individual was tested by their own personalized system. Finally, the usability of the proposed system for recognizing gait events was confirmed by applying it for detecting FOG in medicine. However, direct comparison of the presented results with other related work is difficult because of the unavailability of datasets alongside papers. Therefore, we offer our datasets, provided in Appendix A of this paper, for potential future comparison tests (see Appendix A).

The main novelties of the proposed system for recognizing gait events can be summarized as follows:-developing a robust and lightweight gait motion data acquisition system with semi-automatic data labeling,-identifying the attention-based CNNA + RNN supervised ML classification algorithm as the most reliable method for detecting specific human gait events, justified by comparative analysis with other ML methods,-comparing commonly used ML algorithms with the CNNA + RNN method according to five evaluation metrics, and-confirming the general usability of the system by detecting FOG in medicine.

The rest of the paper is organized as follows. Section 2 describes the hardware materials used, the gait motion data acquisition system and its development, and gives an overview of common ML methods and evaluation metrics, as well as a detailed description of the dataset recording protocol. The results for each tested method, with classification, visualization and evaluation results, are presented in Section 3. The reliable gait analysis results were obtained for healthy subjects, and furthermore, were tested on a patient with a gait disorder. The proposed approach for gait analysis enabled successful FOG detection in PD patients, which is discussed in Section 4, along with the discussion and future work comments.

## 2. Materials and Methods

An overview of the wearable IMU devices and ML methods served as a good starting point for the development of reliable and robust methods for gait analysis. The paper proposes a wearable system, which consists of a data collection system, a data labeling system, a Personal Computer Radio Frequency (PC-RF) transmitter and an elastic strip equipped with IMU and strain gauge sensors. Various ML approaches were applied and tested searching for a proper data processing method. The tested ML methods were evaluated with predefined metrics of accuracy, precision, sensitivity, specificity, F1-score, and Standard Deviation [45]. 

### 2.1. Materials

Two MPU6050 IMU sensors (a GY-521 breakout board) were purchased from 3Dsvet s.p. (Ljubljana, Slovenia). Each sensor contained a three-axis accelerometer (±2 g acceleration scale) and a three-axis gyroscope (±250 °/s angular velocity scale). Strain gauge sensors, along with a 24-bit ADC, were purchased from Conrad (Ljubljana, Slovenia). The sensors utilized i2c communication, requiring 4 wire conductors. A copper coil wire with a thickness of 0.25 mm was purchased from TME GmbH (Łódź, Poland), and used to connect the sensors to the microcontroller. The STM32F103 microcontrollers (32-bit, 72 MHz, 64 KB flash) were purchased from SemafElectronics GmbH (Vienna, Austria). The HC-12 RF modules were purchased from TinyTronics b.v. (Eindhoven, The Netherlands). The INR 35F 18650 3000 mAh Li-Poly batteries were purchased from HTE d.o.o. (Maribor, Slovenia), and used to power the data collection and data labeling systems autonomously. A single cell Li-Poly 3 A Battery Management System (BMS) was purchased from Open Electronics U.A.B. (Kėdainiai, Lithuania) and utilized for safe battery charge and discharge cycles.

### 2.2. Gait Motion Data Acquisition System

A robust gait motion data acquisition system was developed, allowing for the measurement of human limb acceleration, angular velocity and muscle activity throughout time. The proposed system consists of four subsystems (Figure 3):sensors integrated into an elastic strip,a data collection system,a data labeling system, anda PC-RF transmitter.

#### 2.2.1. Sensors Integrated into an Elastic Strip

The IMU sensors were attached to a 50 mm wide elastic strip that can be secured to any human limb to measure the acceleration and angular velocity of it. This study was focused on gait motion analysis, meaning that the sensors needed to be placed somewhere on the lower limbs, in order to be exposed to the highest amount of inertial forces. Different sensor locations were tested (e.g., on the ankle, above the knee and below the knee). Sensor placement below the knee joint, anterior of the shin bone offered the most distinction in the acquired data during different gait activities, which manifested in higher performance of the classifier algorithm. The MPU6050 IMU sensors were considered because they are small (20 × 15 mm) and can be mounted easily to human limbs, and they also consume a tiny amount of power (12 mW), but, at the same time, offer the most amount of data diversity, since different gait activities involve different inertial forces acting on the sensors. Later, strain gauge sensors were integrated into an elastic strip as well. The strain gauge sensor was glued to a 1 mm thin plastic ‘arc’, which flexes under activity from the gastrocnemius leg muscle. Bending of the plastic arc is measured by two strain gauge sensors, wired in a whetstone half-bridge configuration in order to eliminate temperature drift. The strain gauge sensor information is used alongside the IMU sensor information for gait classification.

#### 2.2.2. Data Collection System 

The STM32F103 microcontroller was chosen for all the above mentioned systems, due to its small size (22 × 53 mm), and 32-bit operating nature (0.38 Mflops). The HC-12 RF wireless transmitter modules were chosen for their easy implementation and reliable data transfers up to 10 m. 

The data collection system was miniaturized to achieve an unobtrusive acquisition procedure. Its outside dimensions were 78 × 50 × 25 mm, weighing 120 g, and as such, it can be mounted directly onto a person’s belt via the clip provided, or placed in a pocket (Figure 2). Power consumption of 440 mW was measured (120 mA at 3.7 V). A battery capacity of 3000 mAh is sufficient for 24 h autonomy. A 16 Gb micro-SD card was utilized for data storage. The data collection system was programmed to collect data from the sensors and a data label from the data labeling system and store it on a micro-SD card. 

#### 2.2.3. Data Labeling System

The primary purpose of the data labeling system is creating categorical data label for the acquired data in real-time, by using five separate buttons, where each button press represents a specific predefined gait activity (e.g., standing still, walking, running, Descending Stairs (DS) and Ascending Stairs (AS)). 

The data labeling system is programmed to monitor five button states during gait recording. When one of the buttons is pressed by the user, the system updates the data collection system immediately (via RF communication) with a data label (essentially revealing what gait activity is currently being recorded). The data labeling system’s outside dimensions are 78 × 43 × 26 mm.

#### 2.2.4. PC-RF Transmitter

The fourth part of the proposed gait motion data acquisition system, named a PC-RF transmitter, contains only two components, a microcontroller and an RF module. These enable wireless connection of the data collection system through a serial terminal on the PC. 

The PC-RF transmitter is not essential for the data collection and labeling process, and as such, does not need to be utilized, but is convenient, as it can be used for controlling and monitoring the data acquisition process.

Further data processing, visualization and classification was done in the Matlab 2022a software running on a Windows 10 PC with an i7 7700 processor, 48 Gb RAM and an Nvidia GTX 1070 graphics card. The STM32F103 microcontrollers were programmed in Arduino IDE software utilizing the C programming language.

### 2.3. Methods

The proposed method for human gait activity recognition presents a complex process consisting of the following four components:time-series data collection,data pre-processing,ML algorithms, andmetrics for results evaluation.

In the remainder of this section, the aforementioned components are described in detail.

#### 2.3.1. Time-Series Data Collection

Accelerometers encompass a certain amount of noise in the acquired signal, while gyroscopes are subject to signal drift that increases linearly with time. Sensor fusion methods like the Kalman filter [46] and complementary filter [47] were used in this case. The complementary filter provides us with the ‘best performance of both sensors’. In the short term, gyroscope data was used, because it is exact, and not dependent on external forces, while, in the long term, accelerometer data was used as it does not drift. The fast computation time of the complementary filter enables online operation (calculation at every sample time). A Kalman filter, however, is computationally more complex, and was avoided in this study.

The acquired data from the accelerometer and gyroscope each contain 6 signals (left sensor x, y, z and right sensor x, y, z coordinates), resulting in 12 attributes. An additional 6 signals are generated with the complementary filter, increasing the number of attributes to 18. With the addition of left and right strain gauge sensor information and the data label, we get a total of 21 distinct signals (Figure 4) that are used for further processing. 

The stream of sensor data is stored into a matrix with a rate of 40 Hz (every 25 ms), where each new measurement is stored into a new matrix row. The recorded gait data matrix is visualized (Figure 5), and different gait activities can be observed throughout time.

#### 2.3.2. Data Pre-Processing

In our study, we deal with signals that need to be transformed into features before processing them with ML. The features correspond to signals gathered from sensors. Therefore, characteristics of these features (i.e., attributes) depend on the sensors data, as previously discussed in Section 2.3.1. Mapping from the signals to features together with their characteristics are presented in Table 1, from which it can be seen that there are 20 features in the dataset, to which the corresponding class (i.e., data label) is assigned. All the features, except for data label (class), are characterized by numerical attributes (later normalized to the interval [0, 5]). 

ML algorithms are heavily dependent on the information with which they are trained. The input data must consist of sufficient amounts of information that involves some kind of ‘meaning’ (and not random information, i.e., noise) for the algorithm to learn the desired task properly. 

With data pre-processing, different aspects and parameters of data can be altered, transformed and adjusted significantly, with the goal to create less sparse data that are saturated with useful information. 

The following three data pre-processing methods were utilized:feature normalization,feature transformation, andfeature reduction.

Feature normalization is one of the easiest pre-processing techniques. Normalization is a scaling technique in which values are shifted and rescaled so that they end up ranging between two desired values (0 and 5 in our case—Figure 5). The complementary filter has the smallest amplitudes that could get neglected. Using unscaled variables for training the ML would result in the algorithm preferring variables (sensors) with higher amplitude and neglecting ones with smaller amplitudes. Later, we demonstrate that, for a good gait recognition ML algorithm, feature normalization is all the pre-processing you really need.

Time to frequency domain transformation techniques are often utilized to identify signals underlying frequency components. Frequency transform techniques increase data dimensionality, and often reveal a broader perspective on data. CWT is used commonly in this regard. It measures the similarity between the signal and wavelet analyzing function, with the goal to identify the frequencies and amplitudes that are present in the signal throughout time [42,43,48]. 

Human gait activity generally consists of frequencies between 0.5–10 Hz, therefore a frequency window of 0.2–20 Hz was selected to incorporate all the necessary information (Figure 6). CWT is a continuous 1-D transform function, so the input must be a 1-D single variable vector, from which 27 new frequency features are generated, where each feature corresponds to a specific frequency, with its value indicating its amplitude. A frequency window of 0.2–20 Hz gets divided into 27 smaller ‘frequency windows–features’. CWT was performed for all 20 acquired sensor signals independently, and later combined, generating a matrix containing 540 variables that we used to train the ML algorithms. 

Feature reduction techniques are used to reduce the number of data dimensions without losing important information [37]. With dimensionality reduction the execution time of the ML algorithm is reduced significantly. When the number of dimensions (features) decreases, data become less sparse and more saturated with useful information. Feature reduction can be performed by the so-called AE, which is an unsupervised NN that has the ability to learn efficient data compression and decompression. The weights and biases of the network’s neurons are adjusted to achieve minimal reconstruction loss (comparing input features and reconstructed features). The AE consists of a minimum of 3 layers: An encoder, a bottleneck and a decoder (Figure 7). The bottleneck layer includes a smaller number of neurons (the bottleneck itself), which need to learn the most vital information encompassed in the input features to reconstruct them successfully through the decoding process. Once the AE learns feature reconstruction, only the trained encoder and bottleneck layer are needed for data compression (feature reduction). 

#### 2.3.3. Machine Learning Algorithms

In this study, ML algorithms were used for classification of the recorded and processed data (Figure 5 and Figure 6) into five pre-specified gait activities.

The data label information was used as a ‘target’ for training the ML algorithm (a target indicating the desired output of the algorithm), so that when the trained ML algorithm is tested on an ‘unseen’ gait recording, the algorithm outputs (reveals) different gait activities within it (essentially a trained ML algorithm then creates data labels).

The following nine different ML algorithms were applied, trained, tested and evaluated:the ANN,the DT,the SVM,the NB,the LSTM,the AE + Softmax layer,the AE + biLSTM,the CNN + RNN, andthe CNNA + RNN algorithm.

In the remainder of this chapter, the aforementioned ML algorithms for classification of gait activity are described briefly.

ANN is a commonly used ML algorithm consisting of multiple layers, each consisting of numerous neurons. At its core, each neuron consists of two variables: Weight and bias. An ANN is a complex system that exhibits emergent behavior—the interactions between neurons enable the network to learn [49]. During the learning process, the weights and biases of all neurons are adjusted and optimized until the goal classification error is reached.

DTs are popular, non-complex tools for regression and classification purposes. They have a flowchart like structure which contains a map of possible outcomes of a series of related choices [27]. Usually, DTs start with a single node (root), which, during training, branches into multiple possible outcomes (leaves). It can later classify new instances by passing them down the tree structure from the root to some leaf node, which provides the classification of the instance.

SVM is another popular, powerful, but computationally complex classification method, where an N-dimensional hyper-space is created (N is the number of data dimensions). The data points are ‘plotted’ into hyperspace, and a hyper-plane with a maximum margin (the maximum distance between data points of different classes) is searched for [32]. Hyper-planes represent decision boundaries that can be used to classify new instances.

The NB model is a probabilistic multi-class classifier. Using the Bayes theorem, the conditional probability is calculated for an instance belonging to a class [28]. An assumption is made that the time domain values associated with each class are distributed according to Gaussian distribution when dealing with this data domain. An NB model is computationally undemanding, and is suitable for online adapting (i.e., learning) at every time step on the microcontroller.

LSTM is a type of Recurrent NN (RNN) designed to learn long-term temporal dependencies on time-series data [29]. The neurons in LSTM networks are capable of removing information from the neurons states selectively and enabling the LSTM network to forget certain unnecessary information selectively and sustain its hidden state through time. A normal (unidirectional) LSTM network structure can only learn long-term dependencies from the past, as the only inputs it has seen are from the past. A special bidirectional LSTM structure can process inputs in two ways: One from past to future and another from future to past. In its core, it consists of two individual LSTMs, each wired in a different direction. biLSTMs effectively increase the amount of information in the NN, resulting in better context understanding of the network. Previous research on biLSTM networks in various fields [50,51,52] has shown improvement on time series prediction compared to basic LSTM networks.

An AE with an added softmax layer (output layer) can be used directly in classification tasks. AE reduces the number of dimensions in the data for the softmax layer, which acts as a direct classification layer [53]. The softmax neurons are activated according to a softmax activation function assigning decimal probabilities to each class, allowing prediction of instances. The softmax activation function is calculated using Equation (1):(1)$$\sigma ={\frac{e^{z_{i}}}{\sum_{j=1}^{n}e^{z_{j}}}}_{,}$$
where *σ* is the result of softmax, *z* is an input vector, and the number of classes *n.*

An AE in combination with biLSTM NN improves time series prediction capabilities further [54], as the AE transforms the input data into a less sparse, information-rich format, which is used by the biLSTM NN to learn long-term dependencies. 

A simple CNN + RNN algorithm combines the advantages of both architectures. The Convolutional layers have learnable filter parameters which can learn to produce the most meaningful feature maps for successful classification by later recurrent layers of the algorithm [35,36,37,38,39]. The 1D convolutional layer allows for processing of time-series data, whereas the 2D and 3D convolutional layers are designed for image processing. The CNN + RNN algorithm architecture is presented in Figure 8.

The CNN + RNN algorithm consists of only three convolutional and two fully connected layers wired in series, as shown in Figure 8.

In Deep Learning, the attention mechanism learns to model the importance inside the learning data through optimizing attention weights ‘$a_{w}$’. The attention mechanism learns to enhance the most important aspects of the data, while diminishing other, less important aspects. The architecture of the proposed attention based CNNA + RNN algorithm is presented in Figure 9.

Each stream of CNN layers learns to produce the most meaningful feature map in its unique way. Instead of a single stream CNN + RNN network (Figure 8), parallel CNN streams act as a multi-headed convolutional attention, allowing the latter RNN network to learn, from different feature map representations only, the most consistent features [40,55]. 

The proposed attention mechanism adds up feature maps from multiple parallel streams of CNN layers (Figure 9). This produces a feature map, where similar features of all parallel CNN streams are enhanced, while features that are unique to individual parallel streams are diminished. Visualization of the CNNA + RNN layer activations for gait activity recognition is supplied in the Appendix A.

#### 2.3.4. Metrics for Result Evaluation 

Evaluation metrics play a crucial role in achieving the optimal classifier during the classification training [56]. While the majority of studies addressing ML classification assess performance only by classification accuracy, this does not tell the whole story by itself, since the accuracy correctly considers classified instances only. In line with this, the algorithm’s performance in our study was assessed using six evaluation metrics: Accuracy, precision, sensitivity, specificity, F1-score and Standard Deviation.

The accuracy is the ratio of correctly classified instances according to all instances, and is calculated using the following equation:(2)$$Accuracy={\frac{TP+TN}{TP+TN+FP+FN}}_{,}$$

The true positive (*TP*) and the true negative (*TN*) variables refer to the number of instances correctly classified by the algorithm, while the false positive (*FP*) and the false negative (*FN*) variables refer to incorrectly classified instances. It is desired to maximize the *TP* and *TN* rates while minimizing the *FP* and *FN* rates. The precision is calculated by dividing relevant instances and all retrieved instances in the equation:(3)$$Precision={\frac{TP}{TP+FP}}_{,}$$

The sensitivity (i.e., the *TP* rate) is a ratio of the *TP* among all the relevant instances, and is calculated by the equation:(4)$$Sensitivity={\frac{TP}{TP+FN}}_{,}$$

On the other hand, the specificity (i.e., the *TN* rate) is defined as the ratio of the *TN* amongst all relevant instances, as is formulated in the following equation:(5)$$Specificity={\frac{TN}{TN+FP}}_{,}$$

The F1-score is the harmonic mean of precision and sensitivity, calculated as follows:(6)$$F1=2\times {\frac{precision\times sensitivity}{precision+sensitivity}}_{,}$$

All the aforementioned metrics are referring to single class classification evaluation only. This paper studies a multi-class classification task, where the evaluation metrics are calculated individually for each class, and averaged. In this sense, the Standard Deviation (STD) is defined as the amount of dispersion of a set of values, in other words:(7)$$STD={\sqrt{\frac{\sum_{i=1}^{n}{\left(x_{i}-\tilde{x}\right)}^{2}}{n-1}}}_{,}$$
where *n* is the total number of sample elements, $\tilde{x}$ is the sample mean, and $x_{i}$ refers to the current sample. Low values in Standard Deviation indicate that the set of values tend to be close to its mean value [45].

### 2.4. Dataset Recording Protocol

Informed consent was obtained from all subjects involved in the study. The datasets were recorded according to two protocols that were introduced and are described in detail in the remainder of this chapter. 

#### 2.4.1. Gait Activity Recording Protocol

Subjects participated on a 30 min intense gait exercise at the university building, during which their gait was recorded using the aforementioned gait motion data acquisition system. The original 30 min recordings, which consisted of 72,000 samples, were divided into 10 segments of equal duration, resulting in segments with a duration of 7200 samples (3 min), where each segment must include all five gait activities needed for the classification. No missing instances were identified in the dataset. Gait recording was performed both indoors and outdoors, where each recording took place on a distinct random path, to rule out the possibility of ML ‘preferring’ certain paths with larger amounts of recorded data. We really wanted to explore the boundaries of our gait datasets and include as many rare-events as possible, such as opening the door, walking through a crowd, slipping, crowded stairs, almost falling, etc. Additionally, subjects were visibly exhausted towards the end of the 30 min recording, which has its own effect on gait symmetry, amplitude, rhythm and tempo.

During the gait recordings, the data labeling system’s buttons were pressed by the user accordingly. Activities were addressed by different buttons in the following manner: Standing still (button 1), walking (button 2), running (button 3), ascending stairs (button 4), and descending stairs (button 5). 

Aforementioned gait activities were unevenly distributed in the datasets due to each subject’s gait being recorded on a unique path. The distribution of a certain gait activity can be determined by dividing the length of that activity by the total length of the gait recording. Table 2 presents the activity distribution for the datasets of five different subjects used in our experimental work. 

Relatively high activity imbalances in datasets can be observed in Table 2. Inspecting the mean values of activity distributions, running was the sparsest activity (at only 6.8%), while 58.8% of the datasets consisted of walking.

#### 2.4.2. PD Patient Recording Protocol

The PD patient’s (Hoehn-Yahr stage 4) recording session was conducted in accordance with the declaration of Helsinki and approved by the RS National Medical Ethics Committee (Approval no. 012052/2020).

The PD patient’s gait was recorded for 24 min at the clinic building. During the gait recording session PD patient was asked to perform everyday tasks e.g., standing still, random walking around the clinic, fast walking, rotating, sitting, opening the door, etc. Additionally, sometimes, during walking, the patient was asked to subtract numbers out loud, to simulate real-world conditions better (by increasing his mental workload). A total of 58,004 samples were collected during the recording session, where a total of 13 FOG episodes (representing 8.2% of the dataset) were identified and labeled by the data labeling system, where ‘1’ represented a FOG episode and ‘0’ represented every other activity.

## 3. Results

The goal of our experimental work was firstly focused on identifying the best performing ML algorithm for gait activity classification of a single 25-year-old male subject. Furthermore, the robustness of four best performing algorithms was evaluated with a larger dataset of five subjects. Finally, a follow-up proof of concept FOG detection was performed on the gait data from one male PD patient who commonly experiences FOG episodes. 

To perform a fair comparison, tuning of parameters was performed until the best classification performance was found for each algorithm (independently on the CWT and RAW datasets). The best parameter settings are presented briefly in Table 3, while the full parameter settings and architectures are supplied in the Appendix A.

Three experiments were conducted as follows:recognizing gait events for one subject,recognizing gait events for five subjects, andPD patient FOG episode’s detection.

The first two experiments dealt with a multi-class classification problem with five outputs, while the third presented a binary classification problem with two outputs. The same ML algorithm parameter settings were used to obtain results for all three aforementioned experiments.

### 3.1. Recognizing Gait Events for One Subject

The purpose of the test was to find the ML algorithm that produced the best classification results for Subject 1 (Table 2), according to standard classification metrics. This algorithm presents the most reliable solution, that is then applied in the experiments which follow. We hypothesized that the identified algorithm could be used to detect any kind of desired gait event from input data accurately, given that the event itself has some kind of specific time domain or frequency domain signature. 

The data label is utilized as ground truth in classification evaluation, where the algorithm’s output and data label are compared, and evaluation metrics are calculated. All the ML algorithms were firstly trained with only one 3 min gait recording (train dataset), and later tested on the remaining 27 min of the gait recording (test dataset) in order to evaluate the trained algorithm. The performance of the particular ML algorithm was evaluated using the leave-one-out cross-validation method, where we repeated the process 10 times (each time training the algorithm with a unique 3 min recording). At first, the experiment was conducted using the RAW dataset, which consists of 20 features, and then repeated for the CWT dataset, which consists of 540 features.

In the remainder of this subsection, the results of the experiments are explained and presented in detail (Table 4, Table 5, Table 6 and Table 7), while their further discussion is summarized in the next section. The ML performance of nine algorithms compared regarding the five evaluation metrics is presented in Table 4. The results are calculated for the CWT and RAW datasets separately. Furthermore, it is desired to use the RAW dataset for classification, since we do not have to calculate CWT at all. The best results according to each classification measure are presented as bold in the tables.

Table 4 identifies the CNNA + RNN method as the best performing on the RAW dataset, while the CNN + RNN algorithm outperformed CNNA + RNN for the CWT dataset. The best precision score on the CWT dataset was achieved by the AE + biLSTM method.

Table 5 presents the calculated Standard Deviation for the cross-validated evaluation metrics. The results were calculated for the CWT and RAW datasets separately.

Independent of the 3 min recording we used for training during the leave-one-out cross validation, the CNN + RNN algorithm operates consistently with minimal variations in performance. This fact can be observed as the low Standard Deviation in Table 5.

Table 6 presents the execution time of the tested ML algorithms. It is divided into two parts: The algorithm’s learning time (specified as the duration of time for the algorithm to train), and the algorithm’s classification time (the duration of the pre-trained algorithm to classify new instances). 

As shown in Table 6, the CNNA + RNN is not the fastest computing algorithm but offers the best classification capabilities with the RAW dataset and a still manageable execution time.

Table 7 presents the results from evaluating the CNN + RNN algorithm after training with different combinations of Complementary Filter (CF), Accelerometer (ACC), Gyroscope (GYRO) and Strain Gauge (SG) RAW sensor information.

Inspecting the results from Table 7, we can observe that some sensor combinations work better than others. The combination of all four sensors still showed superiority, but it is worth noting that using an accelerometer or gyroscope alone still yields good classification results. 

Similar to the plot in Figure 5e, it is now possible to visualize the output of the trained CNNA + RNN ML algorithm, as illustrated in Figure 10, where Figure 10b represents the algorithm’s output on raw test dataset, and different gait activities are recognized within the recordings.

Closely inspecting the algorithm’s output reveals some misclassified instances and some detection time asymmetries after comparing it to the ground truth in Figure 10c. The average detection time delay was measured at 100 ms (i.e., 4 samples). Out of 10,000 samples in this test dataset, the algorithm misclassified a total of 22 samples (0.22%).

### 3.2. Recognizing Gait Events for Five Subjects

The purpose of the experiment was to show that the developed system for recognizing different gait events possesses two additional characteristics: Robustness, and personalization. In line with this, the Subjects 1–5 (Table 2) were tested individually by the four best performing algorithms, chosen by the highest F1-score (CNNA + RNN RAW, CNN + RNN RAW, CNNA + RNN CWT and CNN + RNN CWT). The cross-validated results for each subject are presented in Table 8 and Table 9. 

The CNNA + RNN and CNN + RNN algorithms trained with the RAW dataset produced the results as presented in Table 8, while Table 9 corresponds to the results obtained by the same ML algorithms, but for the CWT dataset. 

Comparing Table 8 and Table 9, the CNNA + RNN approach with the RAW dataset delivered the highest performance among all five subjects. 

As can be seen from Table 9, the CNNA + RNN algorithm, trained with the CWT dataset, performed well, but was inferior compared to the CNNA + RNN’s RAW dataset results (Table 8). The classification performance of the first subject (Male, 25) was better compared to the other subjects, due to better data label distribution amongst the dataset, as presented in Table 2. 

For the RAW dataset, the attention mechanism seems only to increase the performance (F1-score) slightly (0.1%) for the first subject, while the performance of the remaining four subjects was improved by 0.45% on average. These findings indicate that the contribution of attention mechanism is greater when dealing with more sparse data label distribution datasets.

### 3.3. PD Patient FOG Episode’s Detection

The purpose of the experiment was to show the usefulness of the proposed system for detecting special gait events like FOG episodes in medicine as well.

In the later stages of the disease, PD patients experience special FOG episodes, where the patient is not able to move the lower limbs despite the clear intention, during which trembling of the lower limbs is often observed. FOG episodes are defined as rare, brief episodes (1–20 s), and typically occur when the patient’s focus is shifted from the gait itself [57], which manifests in one of two scenarios—from initiating gait (sudden intention to move the lower limbs after standing still) or during walking (through doors and avoiding obstacles). The latter is easier to detect, because it consists of a larger amount of gait motion, which manifests in larger amplitudes gathered from the IMU sensor. On the other hand, the strain gauge sensor information is valuable for detecting an FOG during gait initialization, since there are generally low IMU sensor amplitudes present, but the leg muscles of PD patients still tremble, and subsequently produce large strain gauge sensor amplitudes. Our gait recordings include both of the above-mentioned FOG scenarios, and both were detected correctly. 

The PD patient’s 24 min gait recording was split into a train dataset (60%) and test dataset (40%). The results of the latter are visualized in Figure 11a, alongside the algorithm’s output in Figure 11b. Figure 11c compares the algorithm’s output with the data label. 

Each PD patient experiences unique FOG events, which make it difficult to develop a universal ML algorithm for FOG detection (for multiple subjects). This is where our approach with real-time data label creation excelled, as it allowed for complete algorithm personalization.

The FOG detection results for the four best performing classification algorithms are presented in Table 10.

Training the CNNA + RNN and CNN + RNN algorithms with the CWT frequency matrix produced superior results when compared to training with the RAW dataset, as is shown in Table 10. 

Sensor importance for FOG detection was evaluated and presented in Table 11.

As is evident in Table 11, the CNN + RNN algorithm trained with the combined CWT data, obtained by all four sensors, showed superiority, although it also performed relatively well for some of the other tested sensor combinations, e.g., training the ML algorithm using ACC, GYRO and ACC, CF combinations produced the highest precision and sensitivity scores, respectively.

## 4. Discussion

A feasible system capable of gait activity recognition was developed, tested and assessed in this study. The following characteristics of the system can be exposed after our experimental work:reliability,personalization,usability, androbustness.

With the proposed ML algorithms, we demonstrated that the system is feasible to detect specific gait events reliably. Both hardware (the gait motion data acquisition system) and software (various ML algorithms) solutions are presented. The optimal sensor placement location was determined by evaluation of classification metrics. The CNNA + RNN algorithm’s superiority over other ML algorithms can be identified by observing the classification metrics (Table 4 and Table 5). Moreover, proposed algorithm is still relatively fast to compute (Table 6), and therefore, feasible for running online on the microcontroller. We further demonstrated the robustness of four best performing algorithms by performing gait classification on five subjects. The CNNA + RNN algorithm trained with the RAW dataset again showed superiority according to the evaluation metrics.

Gait motion profiles can differ substantially between different humans, especially if movement disorders are present. Therefore, researchers commonly try to develop general ML algorithms which can be applied to multiple subjects. In contrast, the ML algorithms in our study are trained and tested for each individual subject separately. This way, the algorithms can learn subject specific features and achieve complete personalization with only 3 min of recorded gait. The proposed method of automatic online data labeling turned out to be very useful, as it can be used to label practically any desired gait activity in real time, allowing for algorithm personalization to specific subjects (since we have data labels for the whole gait dataset). 

Moreover, the usability of the best performing algorithms was confirmed in a complex practical application (detection of FOG in PD patients), as can be seen in Section 3.3. We discovered that using the CWT dataset yields a more accurate FOG detection algorithm. The trembling of limbs during FOG seems to be more easily distinguishable (from other activities) using the CWT frequency spectrum than the raw signal itself. The data transformation ability of the CWT analysis really seems to reveal broader data insight in this case. In medicine practice, it is desired to detect FOG episodes among all other recorded activities, as this would enable us to assess the effect of pharmacological and nonpharmacological interventions on the occurrence and characteristics of FOG episodes. 

Figure 11 indicates the robustness of our approach to detect even special events that are a result of neurological decline accurately, which points to a possible unobtrusive ‘wearable assistant device’. All the best performing algorithms tested for FOG detection produced some false-positive instances. It is interesting that all the algorithm’s outputs detected roughly the same false-positive instances, hinting at the possibility that there was an onset of FOG actually present and not labeled correctly by the data labeling system’s operator at that specific instance. There is no drawback by providing cues to the patient during false-positive instances, as the patient can only benefit from it. At the end of the day, the algorithm’s outputs suggest, that there is some similarity between false-positive detected instances and FOG instances. The robustness of our system for recognizing gait events is shown in the fact that the system performed exceptionally for all five subjects without the need to change any ML parameters. This holds true for the developed hardware as well.

The proposed system could be upgraded to enable real-time ML algorithm deployment on the microcontroller. Upon adding a cue feedback system, it would be possible to deliver cues ‘on demand’, when specific events are detected. Furthermore, the existing RF communication can be used to activate cue feedback systems wirelessly in real time. This feature has great potential in sports, robotics, virtual reality, medicine (FOG, rehabilitation), etc. 

For potential future comparison tests, we offer our datasets, which are provided in the Appendix A of this paper (see Appendix A).

## Figures and Tables

**Figure 1 sensors-23-00745-f001:**
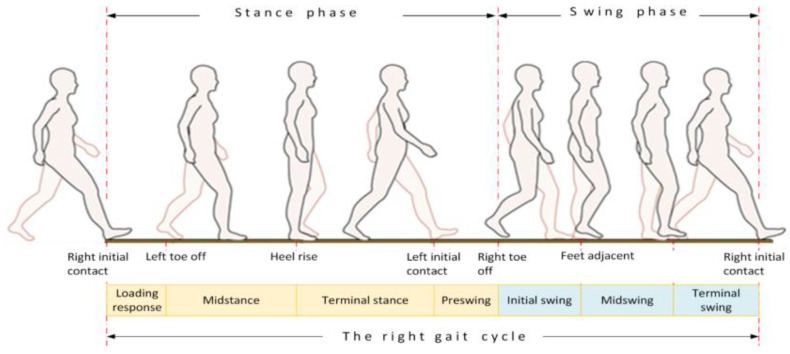
Gait cycle.

**Figure 2 sensors-23-00745-f002:**
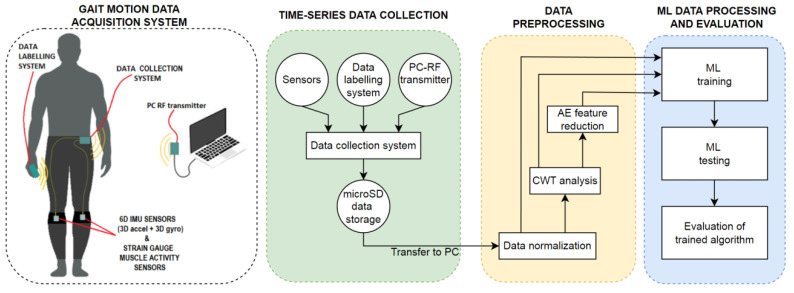
Sketch of the proposed system.

**Figure 3 sensors-23-00745-f003:**
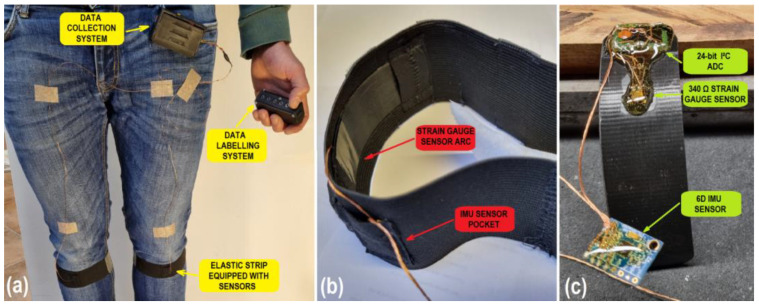
(**a**) Wearing the proposed gait motion data acquisition system; (**b**) close-up picture of the elastic strip equipped with sensors; (**c**) bare sensor unit without the elastic strip. The sensors are covered with elastic glue to increase longevity.

**Figure 4 sensors-23-00745-f004:**
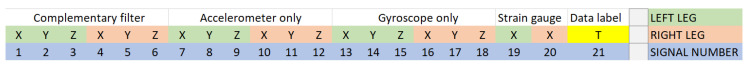
Sensor data structure consisting of 21 signals.

**Figure 5 sensors-23-00745-f005:**
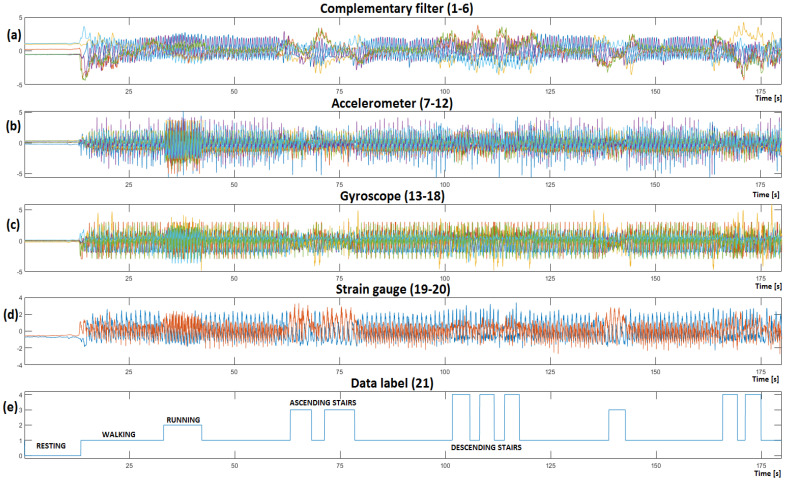
Recorded time–series gait data visualized (the displayed sensor information has already been normalized). Resting can be observed in 0–12 second’s intervals, walking from 12 to 33 s, running from 33 to 40 s, ascending stairs from 67 to 79 s and descending stairs from 105 to 118 s. (**a**) Visualizes the complementary filter’s gait information; (**b**) presents the accelerometer information; (**c**) Shows the gyroscope gait information, while (**d**) visualizes the left and right strain gauge sensor information; (**e**) shows the data label that was created with the data labeling system.

**Figure 6 sensors-23-00745-f006:**
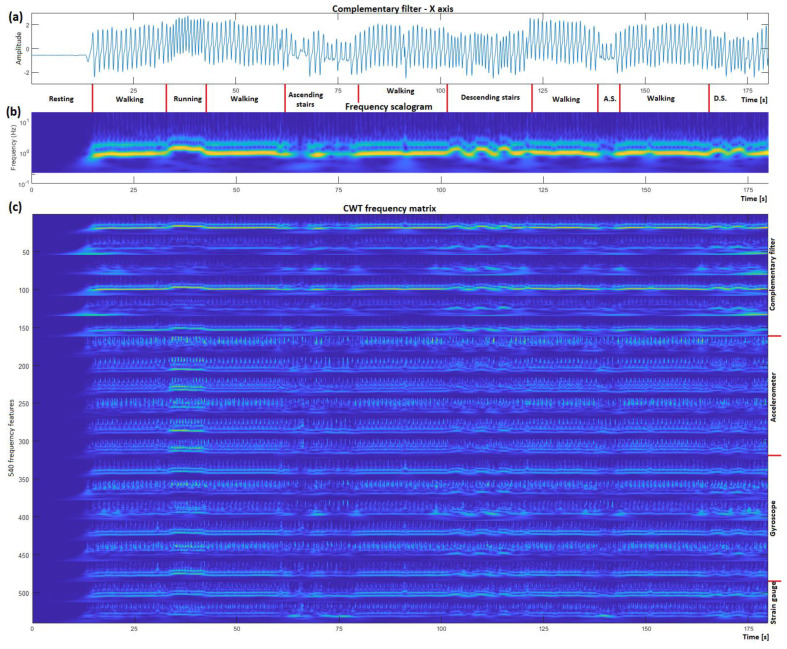
CWT time to frequency domain transformation visualized. (**a**) Presents the normalized X axis of the complementary filter; (**b**) visualizes the result from the CWT algorithm; (**c**) presents the stacked CWT result of each sensor signal.

**Figure 7 sensors-23-00745-f007:**
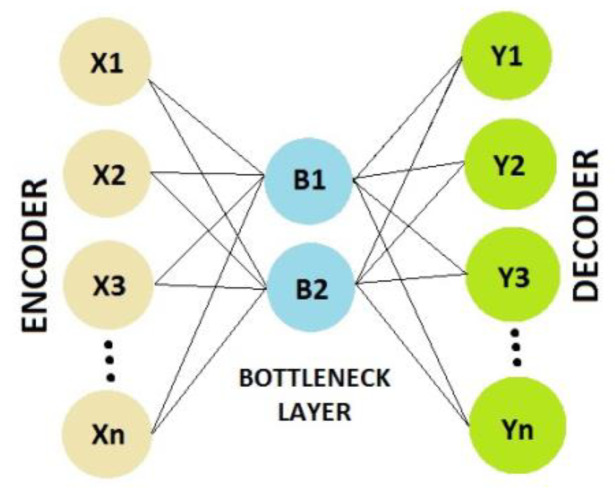
Auto encoder structure.

**Figure 8 sensors-23-00745-f008:**
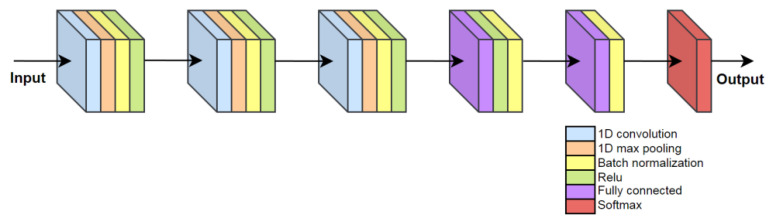
Visualization of the CNN + RNN algorithm architecture.

**Figure 9 sensors-23-00745-f009:**
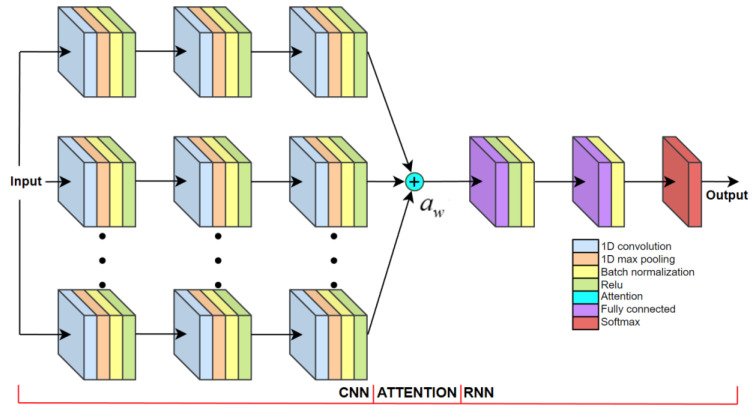
Visualization of the CNNA + RNN algorithm architecture.

**Figure 10 sensors-23-00745-f010:**
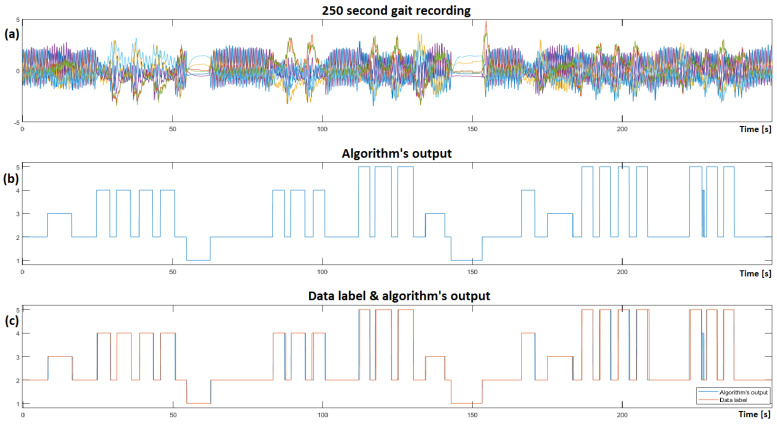
Visualization of the CNNA + RNN algorithm’s output on a test RAW 250 s gait recording; (**a**) Complementary filter information from ‘unseen’ gait dataset visualized; (**b**) illustrates the algorithm’s output on the ‘unseen’ dataset; (**c**) presents a comparison of the data label and algorithm’s output.

**Figure 11 sensors-23-00745-f011:**
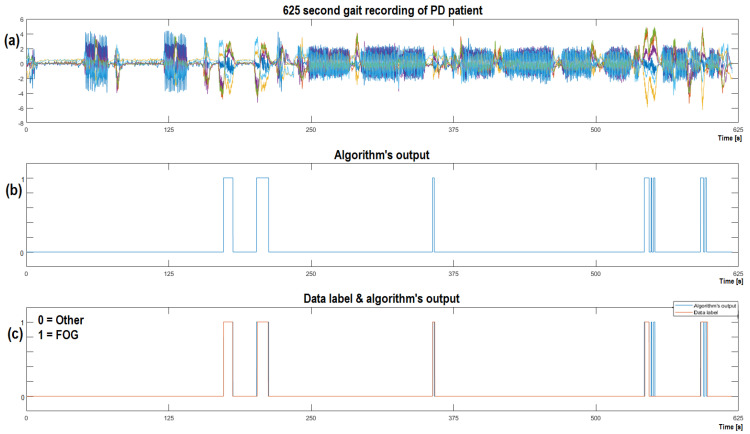
Visualization of the CNNA + RNN algorithm’s output on the test CWT dataset of the PD patient, that experienced five FOG episodes; (**a**) Complementary filter information visualized from an ‘unseen’ gait dataset; (**b**) illustrates the algorithm’s output on the ‘unseen’ dataset; (**c**) presents a comparison of the data label and the algorithm’s output.

**Table 1 sensors-23-00745-t001:** Features and their characteristics.

Sensor	Signals	Position	Features	Attribute Type
Comp. filter	1–6	Left	CF_LEFT_X, CF_LEFT_Y, CF_LEFT_Z	Numeric
		Right	CF_RIGHT_X, CF_RIGHT_Y, CF_RIGHT_Z	Numeric
Accelerometer	7–12	Left	ACC_LEFT_X, ACC_LEFT_Y, ACC_LEFT_Z	Numeric
		Right	ACC_RIGHT_X, ACC_RIGHT_Y, ACC_RIGHT_Z	Numeric
Gyroscope	13–18	Left	GYRO_LEFT_X, GYRO_LEFT_Y, GYRO_LEFT_Z	Numeric
		Right	GYRO_RIGHT_X, GYRO_RIGHT_Y, GYRO_RIGHT_Z	Numeric
Strain gauge	19–20	Left	STRAIN_GAUGE_LEFT_X	Numeric
		Right	STRAIN_GAUGE_RIGHT_X	Numeric

**Table 2 sensors-23-00745-t002:** Dataset activity distribution.

Subjects			Activity Distribution [%]				
Subject	Gender	Age	Resting	Walking	Running	AS	DS
1	M	25	12.2	51.4	9.6	14.4	12.4
2	F	19	9.5	60.5	5.9	13.3	10.8
3	F	60	7.3	61.5	6.3	13.7	11.2
4	M	23	10.4	58.9	6.2	13.5	11.0
5	M	61	8.0	61.8	5.9	13.4	10.9
		Mean	9.5	58.8	6.8	13.6	11.3

**Table 3 sensors-23-00745-t003:** ML algorithm parameters.

Classifier Algorithm	Algorithm’s Hyper-Parameters for the CWT Dataset	Algorithm’s Hyper-Parameters for the RAW Dataset
ANN	Number of learnable parameters = 8.4 k,	Number of learnable parameters = 463,
	Max epoch = 50,	Max epoch = 50,
	Learn Rate = 0.01.	Learn Rate = 0.01.
DT	Number of learnable parameters = 2,	Number of learnable parameters = 2,
	Max. epochs = 50,	Max. epochs = 50,
	Max. Num. splits = 10,	Max. Num. splits = 10,
	Min. Parent size = 2,	Min. Leaf size = 1,
	Num. grid divisions = 10.	Min. Parent size = 2.
SVM	Number of learnable parameters = 6,	Number of learnable parameters = 6,
	Max epoch = 7.	Max epoch = 7.
NB model	Data distribution = ‘Gaussian’,	Data distribution = ‘Gaussian’,
	KFold = 10.	KFold = 10.
LSTM NN	Number of learnable parameters = 410 k,	Number of learnable parameters = 6.9 k
	Max epoch = 60,	Max epoch = 60,
	Learn rate = 0.05.	Learn Rate = 0.05.
AE + Softmax	Number of learnable parameters = 109 k,	Number of learnable parameters = 522,
	Number of bottleneck neurons = 100,	Number of bottleneck neurons = 12,
	Number of softmax neurons = 5,	Number of softmax neurons = 5,
	Encoder Max epoch = 400,	Encoder Max epoch = 400,
	Softmax Max epoch = 400.	Softmax Max epoch = 400.
AE + biLSTM NN	Number of learnable parameters = 171 k,	Number of learnable parameters = 60 k,
	Encoder bottleneck neurons = 100,	Encoder bottleneck neurons = 12,
	Encoder Max epoch = 400,	Encoder Max epoch = 400,
	Max epoch = 75,	Max epoch = 75,
	Learn Rate = 0.05.	Learn Rate = 0.05.
CNN + RNN	Num. of learnable parameters = 9.4 M,	Num. of learnable parameters = 15 k,
	Max epoch = 50,	Max epoch = 50,
	Learn Rate = 0.01.	Learn Rate = 0.01.
CNNA + RNN	Num. of learnable parameters = 47 M,	Num. of learnable parameters = 75 k,
	Parallel CNN streams = 5,	Parallel CNN streams = 5,
	Max epoch = 50,	Max epoch = 50,
	Learn Rate = 0.01.	Learn Rate = 0.01.

**Table 4 sensors-23-00745-t004:** Averaged cross-validated evaluation metrics for the corresponding nine classifiers.

Evaluation Metrics	CWT Dataset					RAW Dataset				
Classifier	Accuracy	Precision	Sensitivity	Specificity	F1	Accuracy	Precision	Sensitivity	Specificity	F1
ANN	0.983	0.968	0.949	0.986	0.958	0.943	0.839	0.863	0.958	0.851
DT	0.957	0.703	0.892	0.968	0.786	0.919	0.652	0.772	0.934	0.707
SVM	0.980	0.957	0.937	0.982	0.947	0.969	0.932	0.912	0.974	0.922
NB model	0.969	0.939	0.905	0.974	0.921	0.901	0.763	0.756	0.923	0.760
LSTM NN	0.930	0.843	0.808	0.945	0.825	0.981	0.955	0.946	0.985	0.951
AE + Softmax	0.980	0.953	0.945	0.984	0.949	0.962	0.910	0.895	0.969	0.903
AE + biLSTM	0.985	**0.972**	0.952	0.986	0.962	0.907	0.783	0.735	0.928	0.757
CNN + RNN	**0.989**	0.97	**0.975**	**0.991**	**0.973**	0.991	0.976	0.983	0.994	0.980
CNNA + RNN	0.988	0.968	0.973	0.991	0.971	**0.992**	**0.978**	**0.983**	**0.994**	**0.981**

**Table 5 sensors-23-00745-t005:** Standard deviation of cross-validated results.

Standard Deviation [×10^−3^]		CWT Dataset				RAW Dataset				
Classifier	Accuracy	Precision	Sensitivity	Specificity	F1	Accuracy	Precision	Sensitivity	Specificity	F1
ANN	3.3	9.4	14.1	3.5	7.8	10.1	26.3	32.0	8.8	28.7
DT	15.0	37.6	20.6	7.6	30.3	15.5	21.2	52.5	14.1	34.2
SVM	9.4	18.3	29.8	8.2	23.4	18.8	38.8	51.8	16.0	45.3
NB model	7.5	9.5	26.9	8.0	16.3	9.0	23.4	33.3	8.0	26.2
LSTM NN	59.5	128.6	148.4	47.5	138.6	6.9	18.9	23.7	4.9	21.1
AE + Softmax	3.8	11.8	13.0	3.1	9.8	5.7	10.3	20.7	5.3	14.7
AE + biLSTM	2.5	11.9	11.1	2.2	6.7	27.9	51.5	91.3	22.8	70.1
CNN + RNN	**1.5**	7.2	7.8	**1.5**	**4.0**	**1.8**	**6.1**	**4.9**	**1.3**	**4.2**
CNNA + RNN	1.8	**6.9**	**7.5**	1.8	4.4	2.1	7.4	5.8	1.3	5.8

**Table 6 sensors-23-00745-t006:** Execution time of the classification algorithms.

	CWT Dataset		RAW Dataset	
Classifier	Learn Time [s]	Classification Time [s]	Learn Time [s]	Classification Time [s]
ANN	27.83	0.47	5.51	0.18
DT	20.78	**0.14**	37.94	**0.03**
SVM	337.62	4.68	238.39	0.67
NB model	**0.25**	1.58	**0.01**	0.06
LSTM NN	224.17	7.66	211.39	7.30
AE + Softmax	158.27	0.64	30.05	0.14
AE + biLSTM NN	197.57	6.62	148.94	6.78
CNN + RNN	47.28	0.71	5.12	0.05
CNNA + RNN	136.64	1.98	17.79	0.20

**Table 7 sensors-23-00745-t007:** Sensor importance evaluation.

	RAW Dataset Sensor Evaluation				
Sensor Combination	Accuracy	Precision	Sensitivity	Specificity	F1
CF	0.971	0.945	0.903	0.974	0.924
ACC	0.989	0.967	0.974	0.992	0.970
GYRO	0.987	0.975	0.965	0.989	0.970
SG	0.978	0.939	0.941	0.984	0.940
ACC, GYRO	0.990	0.977	0.974	0.995	0.976
ACC, CF	0.990	0.976	0.975	0.992	0.975
ACC, SG	0.990	0.974	0.978	0.992	0.976
GYRO, CF	0.983	0.963	0.950	0.985	0.957
GYRO, SG	0.989	0.975	0.972	0.992	0.974
CF, SG	0.983	0.967	0.945	0.985	0.956
CF, SG, GYRO	0.991	0.981	0.977	0.993	0.979
CF, SG, ACC	0.990	0.980	0.972	0.991	0.976
SG, ACC, GYRO	0.991	0.976	0.978	0.993	0.977
CF, ACC, GYRO	0.991	0.978	0.979	0.993	0.978
CF, ACC, GYRO, SG	**0.992**	**0.982**	**0.981**	**0.994**	**0.981**

**Table 8 sensors-23-00745-t008:** RAW dataset evaluation.

	CNNA + RNN					CNN + RNN				
Subject	Accuracy	Precision	Sensitivity	Specificity	F1	Accuracy	Precision	Sensitivity	Specificity	F1
1	**0.992**	**0.978**	**0.983**	**0.994**	**0.981**	**0.991**	**0.976**	**0.983**	**0.994**	**0.980**
2	0.988	0.962	0.973	0.990	0.967	0.986	0.956	0.970	0.989	0.963
3	0.989	0.963	0.979	0.992	0.971	0.988	0.960	0.974	0.990	0.967
4	0.990	0.973	0.975	0.991	0.974	0.989	0.966	0.973	0.991	0.969
5	0.990	0.968	0.980	0.992	0.974	0.989	0.961	0.978	0.991	0.969
Mean	0.989	0.968	0.978	0.991	0.973	0.988	0.963	0.975	0.991	0.970

**Table 9 sensors-23-00745-t009:** CWT dataset evaluation.

	CNNA + RNN					CNN + RNN				
Subject	Accuracy	Precision	Sensitivity	Specificity	F1	Accuracy	Precision	Sensitivity	Specificity	F1
1	**0.988**	**0.968**	**0.973**	**0.991**	**0.971**	**0.989**	**0.970**	**0.975**	**0.991**	**0.973**
2	0.983	0.955	0.952	0.985	0.953	0.984	0.956	0.959	0.986	0.957
3	0.984	0.954	0.959	0.986	0.956	0.983	0.952	0.954	0.986	0.953
4	0.986	0.954	0.963	0.989	0.958	0.984	0.949	0.955	0.988	0.952
5	0.984	0.955	0.962	0.986	0.958	0.985	0.955	0.962	0.986	0.958
Mean	0.985	0.957	0.961	0.987	0.959	0.985	0.956	0.961	0.987	0.958

**Table 10 sensors-23-00745-t010:** Evaluation metrics for four classifiers.

	FOG Detection Evaluation Metrics			
Classifier	Accuracy	Precision	Sensitivity	F1
CNNA + RNN RAW	0.981	0.867	0.964	0.913
CNN + RNN RAW	0.979	0.873	0.917	0.894
CNNA + RNN CWT	**0.988**	**0.929**	0.949	**0.939**
CNN + RNN CWT	0.983	0.872	**0.986**	0.926

**Table 11 sensors-23-00745-t011:** Sensor importance evaluation for FOG detection.

	CWT Dataset Sensor Evaluation			
Sensor Combination	Accuracy	Precision	Sensitivity	F1
CF	0.975	0.843	0.928	0.884
ACC	0.977	0.850	0.944	0.894
GYRO	0.981	0.901	0.884	0.892
SG	0.972	0.817	0.967	0.885
ACC, GYRO	0.983	**0.911**	0.901	0.906
ACC, CF	0.982	0.869	**0.971**	0.917
ACC, SG	0.977	0.855	0.923	0.888
GYRO, CF	0.980	0.892	0.893	0.893
GYRO, SG	0.983	0.903	0.921	0.912
CF, SG	0.980	0.874	0.920	0.897
CF, SG, GYRO	0.980	0.860	0.970	0.912
CF, SG, ACC	0.981	0.862	0.982	0.918
SG, ACC, GYRO	0.983	0.903	0.921	0.912
CF, ACC, GYRO	0.980	0.870	0.946	0.906
CF, ACC, GYRO, SG	**0.986**	0.898	0.964	**0.929**

## Data Availability

The data presented in this study are available in Appendix A.

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
