# Peer review of "Human Gait Activity Recognition Machine Learning Methods"

_sensors, 2023, doi:10.3390/s23020745_

Round 1

Reviewer 1 Report (Previous Reviewer 1)

In the paper, the Authors used several AI algorithms to classify five pre-specified gait activities. The algorithms were tested on 5 healthy subjects and 1 person with PD. However, the dimensions and the characteristics of the dataset used to train the algorithms (before testing them) are unclear. The authors should better describe this aspect.

Author Response

Please see the attachment. We have included response from other reviewer as well.

Reviewer 2 Report (Previous Reviewer 2)

This version was significantly improved.  I think this version can be accepted for publication.

Author Response

Please see the attachment. We have included response from other reviewer as well.

Round 2

Reviewer 1 Report (Previous Reviewer 1)

Thank you. I have no further suggestions.

This manuscript is a resubmission of an earlier submission. The following is a list of the peer review reports and author responses from that submission.

Round 1

Reviewer 1 Report

The study is interesting. However, I suggest to increase the number of subjects (both healthy subjects and patients) to further demonstrate the robustness of the proposed approach.

Reviewer 2 Report

The authors proposed gait activity recognition system allowing the identification of common risk factors and enhancing the subject’s quality of life. Gait movement information was acquired using accelerometers and gyroscopes mounted on lower limbs, where the sensors are exposed to inertial forces during gait movement. Various machine learning methods were tested, with the goal to process the recorded gait information and to identify underlying gait activities. The combination of Auto encoder and the biLSTM neural network algorithm outperformed other tested algorithms and successfully detected all the underlying activities with 98.4% classification accuracy, 96% precision, 94.7% sensitivity, 98.5% specificity and 95.3% F1-score.

The paper needs more m modifications, as follows:

- One of the main problem of those applications is the generality, how did you gurantee the generality of the proposed gait recognition system with different environments and people with different sizes and shapes?

-Clarify the training and testing, also consider the generality problem as mentioned above.

- Clarify the steps of the whole recognition system, add more description.  

 -Improve the quality of some figures, such as Figure 4.

- The parameter settings of all compared deep learning models must be described. How did you guarantee fair comparisons.

- Attention mechanism for deep learning models could be discussed, such as Cjam: Convolutional neural network joint attention mechanism in gait recognition; Multi-ResAtt: Multilevel Residual Network with Attention for Human Activity Recognition Using Wearable Sensors; A lightweight attention-based CNN model for efficient gait recognition with wearable IMU sensors;

Reviewer 3 Report

Authors proposed a IMU-based sensor system embedded with data analysis software based on machine learning technique to be used in gait analysis. The topic is in line with the journal aims. However, the quality of the paper and the significance of contents is too low to be published in an important journal like sensors. The main issue is related to the novelty of the paper. Even though the authors tried to report the novelties of the paper, the four points highlighted in the Introduction do not represent a concrete novelty. The literature is full of "new" sensors applied on gait analysis fields by also using the artificial intelligence. Also the metrics used to test the ML algorithms have been already introduced. In my opinion, authors should report a concrete application of their wearable sensor and find the novelty in the application.

Other issues:

- New sensors should be validated before the application.

- Many information about ML algorithms

- Dimensions of sensors and data acquisition unit should be reported in order to better understand the wearability.